# Humans disrupt access to prey for large African carnivores

Kirby L Mills*, Nyeema C Harris*

Applied Wildlife Ecology Lab, Ecology and Evolutionary Biology Department, University of Michigan, Ann Arbor, United States

**Abstract** Wildlife respond to human presence by adjusting their temporal niche, possibly modifying encounter rates among species and trophic dynamics that structure communities. We assessed wildlife diel activity responses to human presence and consequential changes in predator-prey overlap using 11,111 detections of 3 large carnivores and 11 ungulates across 21,430 camera trap-nights in West Africa. Over two-thirds of species exhibited diel responses to mainly diurnal human presence, with ungulate nocturnal activity increasing by 7.1%. Rather than traditional pairwise predator-prey diel comparisons, we considered spatiotemporally explicit predator access to several prey resources to evaluate community-level trophic responses to human presence. Although leopard prey access was not affected by humans, lion and spotted hyena access to three prey species significantly increased when prey increased their nocturnal activity to avoid humans. Human presence considerably influenced the composition of available prey, with implications for prey selection, demonstrating how humans perturb ecological processes via behavioral modifications.

## Introduction

Wildlife can adaptively respond to their environment by modifying their diel activity and partitioning time to maximize survival and limit exposure to risks, producing a species' temporal niche (*Bennie et al., 2014*; *Vinne et al., 2019*). Prey commonly employ predator avoidance strategies along the temporal niche axis (*Kohl et al., 2019*), which is contrasted by predators selecting for temporal activity patterns that maximize hunting success and minimize competitive encounters (*Cozzi et al., 2012*; *Dröge et al., 2017*). As a result, large carnivores are predominantly nocturnal while ungulates often exhibit more diurnal behavior, although neither exclusively so. However, pervasive human pressures disrupt individual behaviors that facilitate coexistence of predator and prey populations alike (*Wolf and Ripple, 2016*; *Shamoon et al., 2018*; *Xiao et al., 2018*; *Sévêque et al., 2020*). How human-induced responses of many species cascade to alter the dynamics of predation and other ecological interactions at the community level remains understudied (*Guiden et al., 2019*).

The fear of humans can suppress spatiotemporal activity in both carnivores and herbivores with cascading impacts to lower trophic levels (*Dorresteijn et al., 2015*; *Gaynor et al., 2018*; *Suraci et al., 2019a*). Specifically, human presence engenders shifts in diel activity patterns across guilds, altering their temporal niche to incorporate avoidance of human encounters (*Gaynor et al., 2018*; *Frey et al., 2020*). Human activities concentrated in the day and predator activity at night reduce the availability of temporal refugia for prey from risky encounters, and can constrain species' abilities to optimize activity along the temporal niche axis (*Kohl et al., 2019*; *Vinne et al., 2019*). As predator and prey species alter their diel activity to adaptively respond to human presence, predator-prey temporal overlap and resulting encounter rates are likely to be changed (*Patten et al., 2019*), thus altering predator access to a suite of prey resources (*Figure 1*). Such perturbations to predator-prey dynamics can have cascading impacts that alter population regulation, habitat

*For correspondence:
kimills@umich.edu (KLM);
nyeema@umich.edu (NCH)

**Competing interests:** The authors declare that no competing interests exist.

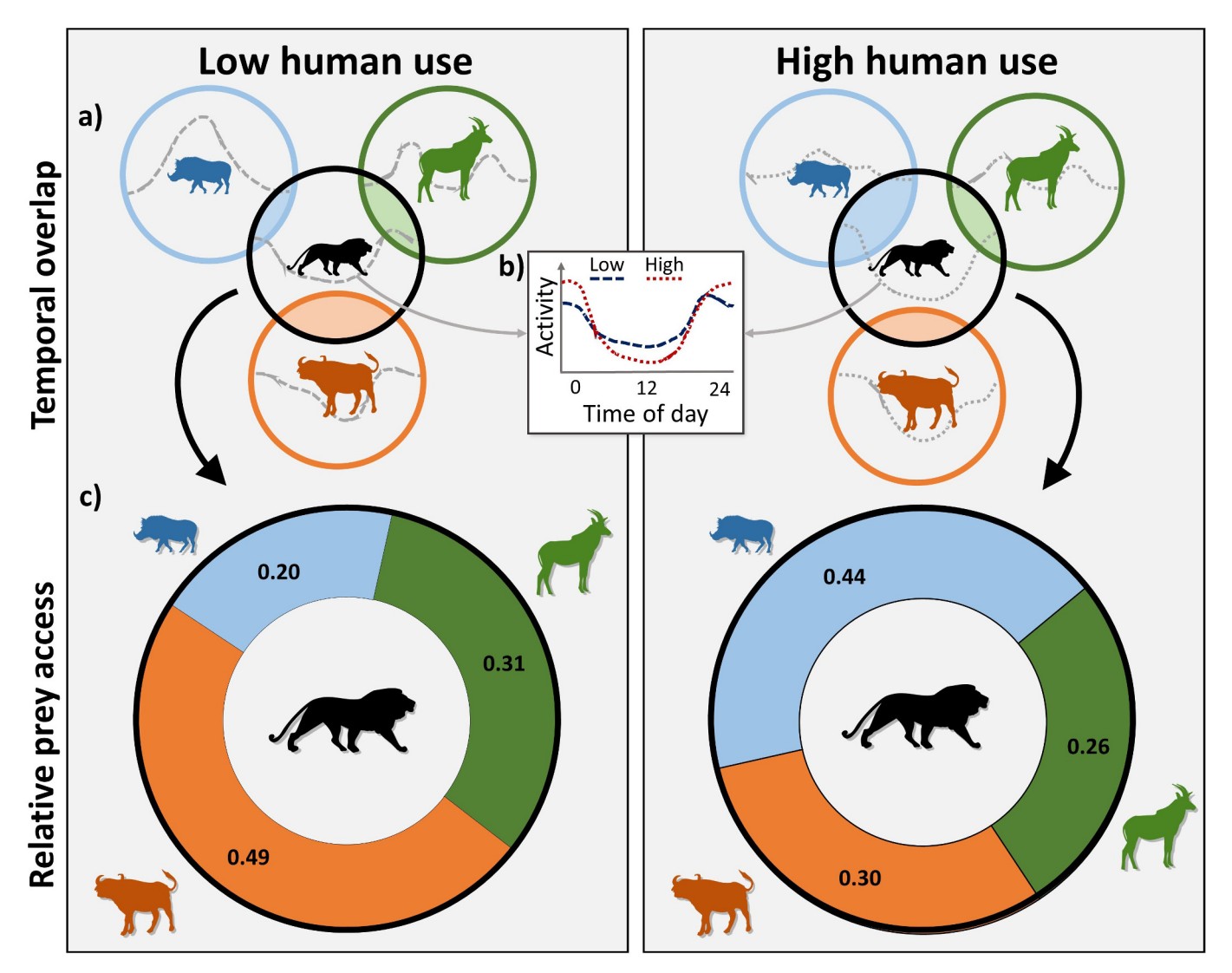

**Figure 1.** Conceptual framework illustrating the community-level effects of human presence on predator-prey temporal interactions. (**a**) Circles represent the temporal niche occupied by each species (three prey and one predator), and shaded regions indicate temporal overlap between the predator and a prey species (i.e. shared temporal niche space). Dotted lines within each circle depict the species' temporal activity distribution. (**b**) The diel activity patterns of both predators and prey are expected to shift in response to human presence, generally increasing nocturnal activity to avoid humans during the day. (**c**) As wildlife diel activity changes, so does predator access to individual prey. Human-induced shifts can lead to intensified or relaxed predation pressures on an individual prey species depending on the diel responses of the prey compared to the predator and other sympatric prey.

structure, and various ecosystem processes, such as carbon storage, herbivory, and seed dispersal (*Pringle et al., 2007*; *Terborgh et al., 2008*; *Asner et al., 2009*; *Schmitz et al., 2018*; *Atkins et al., 2019*).

If wildlife modify their temporal niche to avoid pressures associated with human presence, predators and prey will exhibit increased nocturnal activity at both the species and guild levels (*Gaynor et al., 2018*). If all species respond to humans similarly, human avoidance further predicts: (1) intensified predator-prey overlap overall and (2) a greater diversity of prey species available to predators as previously diurnal species adopt nocturnal behaviors. Increasing the diversity of accessible prey would likely result in diminished predation rates on individual species, given that prey selection by carnivores is influenced in part by prey species' availability relative to other sympatric prey and the diversity of the prey community (*Sinclair et al., 2003*; *Owen-Smith and Mills, 2008*).

However, avoidance of humans may not be ubiquitous across species given that species have different vulnerabilities to humans (*Tablado and Jenni, 2017*). Thus, the prevalence of human avoidance among species is likely to determine the nature of community-level predator-prey outcomes.

Here, we evaluated the effects of human presence on the diel activity of predators and prey and consequential differences in predator-prey relationships using a novel method to assess predator-prey overlap at a community scale. We executed a systematic camera survey spanning 13,100 km$^2$ of the W-Arly-Pendjari (WAP) complex in West Africa across 21,430 trap-nights, obtaining detections of both wildlife and humans. We used occupancy modeling to determine areas of low and high human use within the study area and evaluate spatially explicit responses in species' behavior and potential alterations to trophic interactions. Specifically, we tested for differences in diel activity patterns and nocturnal behaviors for 3 large carnivores (African lions, spotted hyenas, and African leopards) and 11 ungulate species between areas of low and high human presence. We also evaluated the effects of human presence on the overall temporal overlap ($\Delta$) between each predator and its prey, as well as assessed differences in the relative overlap between predators and each individual prey species. We determined: (i) how carnivores and ungulates adjusted their temporal niche in response to human presence and (ii) how apex predator access to prey species was influenced by human presence.

Previous works often investigate temporal overlap of predators and prey in a pairwise manner (*Linkie and Ridout, 2011*; *Ramesh et al., 2012*; *Patten et al., 2019*). However, such an approach does not consider the overall composition of resources available to predators and the relative contributions of individual prey species. Higher order interactions beyond pairwise predator-prey relationships likely contribute to determining community structure and coexistence among species (*Levine et al., 2017*). We combatted these limitations by extending beyond pairwise comparisons to consider predator-prey interactions at the community level. We aggregated temporal activity among ungulates, providing a more ecologically realistic depiction of overlap between predators and their prey. Specifically, we used bootstrapped kernel density distributions of predator and prey diel activity to calculate the overlap between each predator-prey pair relative to the overall available prey (percent area under the predator diel curve, PAUC). PAUC values were generated by aggregating prey activity curves and then scaling the prey activity (kernel density estimates) to each predator. PAUC represents a metric of relative prey access for the apex predator, as it provides insight into the times of day that encounters between the predator and prey species are most likely to occur based on the temporal activity of both. In this new approach to assess the spatially explicit temporal responses of predators and their prey to humans, we elucidate the community-level effects of humans on trophic interactions and their implications for ecosystem regulation by large carnivores.

## Results

Our camera survey yielded 786 and 10,325 detections of apex predators and ungulates, respectively, over 21,430 trap-nights throughout our West African study system (*Supplementary file 1*). Spotted hyenas are the dominant predator in the system with six times more detections than either African lions or leopards. Warthog, reedbuck, and bushbuck were the most commonly observed ungulates, each detected over 1000 times.

We obtained 350 detections of humans in 69 out of 204 surveyed 10-km$^2$ grid cells, leading to a naive human occupancy of 0.34. Accounting for imperfect detection, model selection resulted in four competing top models ($\Delta$AICc < 2) for human occupancy (*Supplementary file 2*). Detection of humans primarily varied among years and sites and was higher in non-savanna habitat (top model goodness-of-fit *p-value* = 0.327). Human occupancy was pervasive, but heterogeneous within the study area (*Figure 2a*; $\bar{\Psi}$ = 0.54 SE 0.41), ranging from 0.0006 to 1 with highest frequencies near these extremes (*Figure 2b*). Using the mean value of occupancy as a pressure threshold, we designated 108 of 204 grid cells as having high human use (occupancy > 0.54). Humans exhibited mostly diurnal activity with 80.3% of detections occurring between sunrise and sunset (*Figure 2c*).

### Human avoidance responses

Human presence generated marked modifications in the temporal niches of sympatric wildlife, with both guilds exhibiting human avoidance behaviors overall. Carnivores and ungulates showed

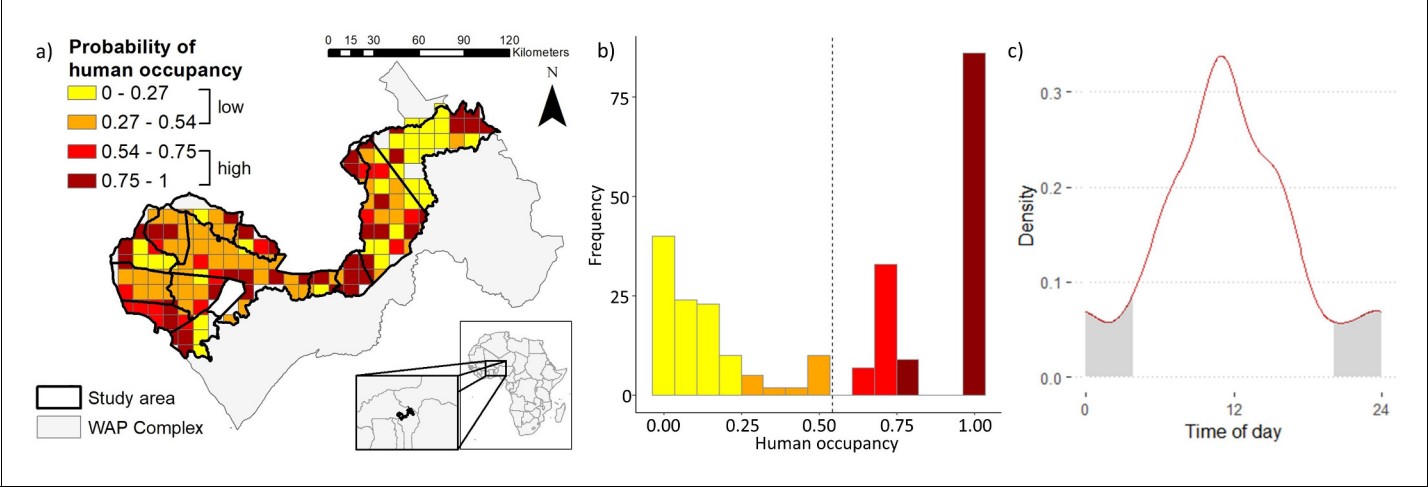

**Figure 2.** Results of spatial and temporal human use within the W-Arly-Pendajri protected area complex in Burkina Faso and Niger, West Africa. (a) Map of study area within the W-Arly-Pendjari complex with surveyed 10-km$^2$ grid cells. Color depicts estimated levels of human occupancy within the study area, averaged across years for grid cells surveyed in multiple years. (b) Corresponding frequencies of grid-level human occupancy for 3 survey years, with dotted line depicting mean human occupancy (0.54). (c) Human diel activity kernel density distribution from camera detections. The online version of this article includes the following figure supplement(s) for figure 2:

**Figure supplement 1.** Camera placement in W-Arly-Pendjari protected area complex from 3 survey years.

significantly different diel activity patterns between low and high human use (carnivores p-value=0.017; ungulates p-value<0.001; *Figure 3*). Over two-thirds (10 out of 14) of the mammal species in the study exhibited significant differences in their diel activity patterns in response to human presence (African leopards, spotted hyenas, and eight ungulates; *Figure 3*). Ungulates overall were 7.1% (95% CI ± 1.7%) more active at night in high human areas, while carnivores showed a slight but non-significant increase in night-time activity of 3.9% (±5.7%). Specifically, we observed significantly higher nocturnal activity with high human use for reedbuck (+12.3 ± 4.8%), duiker (+7.4 ± 4.4%), bushbuck (+6.9 ± 3.6%), and warthog (+4.5 ± 1.9%); and significant decreases for kob (−5.3 ± 4.2%) and aardvark (−15.0 ± 8.1%; *Figure 4*). In contrast, five ungulate species and all three carnivores showed no significant differences in nocturnality. After testing the sensitivity of our results to the human occupancy threshold selected, we found that increasing or decreasing the low vs. high human occupancy threshold by ±0.1 did not alter our interpretation of species' differences in diel activity (*Supplementary file 3*). The only change we observed was detecting significance when reducing the threshold from 0.54 to 0.44 for two species: African leopard and roan antelope. Our results highlight that most species respond to human occurrence by modifying their behaviors and reducing their realized temporal niche to incorporate more night-time activity, potentially altering predator-prey encounter rates.

## Changes in predator-prey overlap

Differences in diel activity among species did not result in significant differences in individual predators' temporal overlap (Δ) with prey when we aggregated their prey species (*Figure 3—figure supplement 1*). However, high human use areas showed lower mean overlap of African lions with their prey by 0.08 ($\bar{\Delta}_{high}$ = 0.718, 95% CI ± 0.08; $\bar{\Delta}_{low}$ = 0.797 ± 0.11). In contrast, African leopards may be experiencing some benefit from human use, as their temporal overlap with prey was 0.17 higher where human activities were high ($\bar{\Delta}_{high}$ = 0.699 ± 0.12; $\bar{\Delta}_{low}$ = 0.529 ± 0.11). Spotted hyenas appear to be robust to human occurrence, showing almost no differences in total overlap with prey due to humans ($\bar{\Delta}_{high}$ = 0.638 ± 0.03; $\bar{\Delta}_{low}$ = 0.625 ± 0.04).

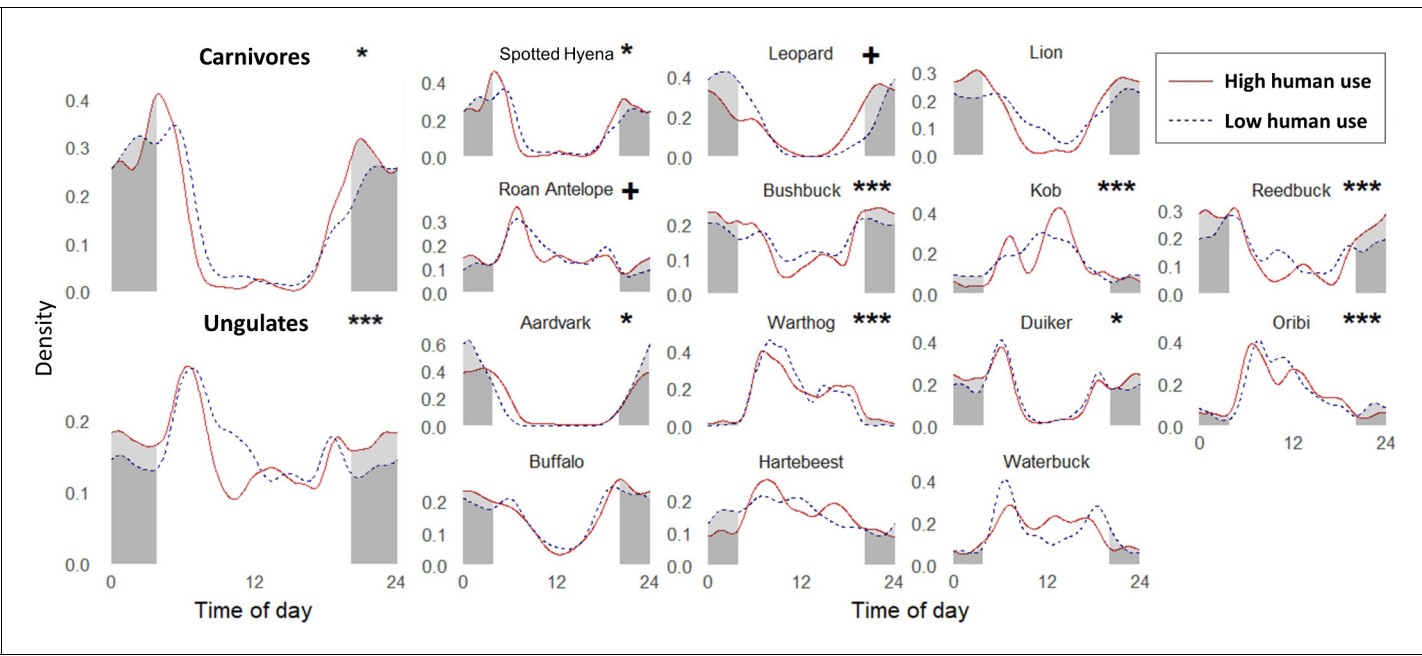

**Figure 3.** Temporal activity kernel density curves for large carnivores (top row) and ungulates in areas of low and high human use (threshold human occupancy = 0.54). Nocturnal diel periods (2 hr after sunset to 2 hr before sunrise) are shaded using the average times of sunrise and sunset during our study period, and lighter shading represents the diel-specific nocturnal activity that is different between low and high human areas. Significance levels for bootstrapped randomization test of differences in diel distributions between human zones: *<0.05, **<0.01, ***<0.001. Plus signs (+) represent species with p-values<0.1 which achieved significance when the human occupancy threshold was adjusted ±0.1 (*Supplementary file 3*). The online version of this article includes the following figure supplement(s) for figure 3:

**Figure supplement 1.** Temporal overlap coefficients (Δ) between each predator and their associated prey species.

## Human occurrence restructures access to specific prey

African lions and spotted hyenas experienced similarly distinct differences in the composition of accessible prey due to human presence using the 95% CIs of the average difference in percent area under the predator activity curve ($\overline{\Delta PAUC}$), our novel method for assessing predator-prey temporal overlap in a community context (*Figure 5a*). Specifically, humans generated significant differences in overlap of these predators with 4 out of 11 prey species: bushbuck ($\overline{\Delta PAUC}_{lion}$ = +1.49, 95% CI ± 1.14%; $\overline{\Delta PAUC}_{hyena}$ = +1.51 ± 0.73%), reedbuck ($\overline{\Delta PAUC}_{lion}$ = +1.99 ± 1.34%; $\overline{\Delta PAUC}_{hyena}$ = +1.88 ± 0.87%), duiker ($\overline{\Delta PAUC}_{lion}$ = +1.56 ± 1.31%; $\overline{\Delta PAUC}_{hyena}$ = +0.84 ± 0.73%), and kob ($\overline{\Delta PAUC}_{lion}$ = -2.24 ± 1.58%; $\overline{\Delta PAUC}_{hyena}$ = -1.45 ± 0.88%) (*Figure 5b*). All three species to which predator access increased significantly also exhibited increased night-time activity as a human avoidance strategy. In contrast, kob was less active at night in high human areas and experienced lower overlap with African lions and spotted hyenas (*Figure 4*). Additionally, African lion and spotted hyena access to two prey species showed near significant differences (buffalo $\overline{\Delta PAUC}_{lion}$ = +1.33 ± 1.41%; $\overline{\Delta PAUC}_{hyena}$ = +0.87 ± 0.97%; and waterbuck $\overline{\Delta PAUC}_{lion}$ = -1.66 ± 1.73%; $\overline{\Delta PAUC}_{hyena}$ = -1.67 ± 1.71%).

All three apex predators showed comparable differences in access to all prey between human use levels (*Figure 5b*). However, differences in African leopard access to prey items were not significant based on 95% CIs, with only aardvark ($\overline{\Delta PAUC}_{leopard}$ = -4.6 ± 4.7%) and bushbuck ($\overline{\Delta PAUC}_{leopard}$ = 1.7 ± 1.8%) access nearing significance (*Figure 5*). We suspect this is due to leopards' differential response to human presence (-4.6 ± 19.1% change in nocturnality) compared to African lions (+11.9 ± 16.3%) and spotted hyenas (+3.7 ± 6.5%; *Figure 4*), although these differences are non-significant.

Although overlap with total available prey did not differ for any predator, human presence increased the variation in species-specific prey accessibility (PAUC estimates) for African lions

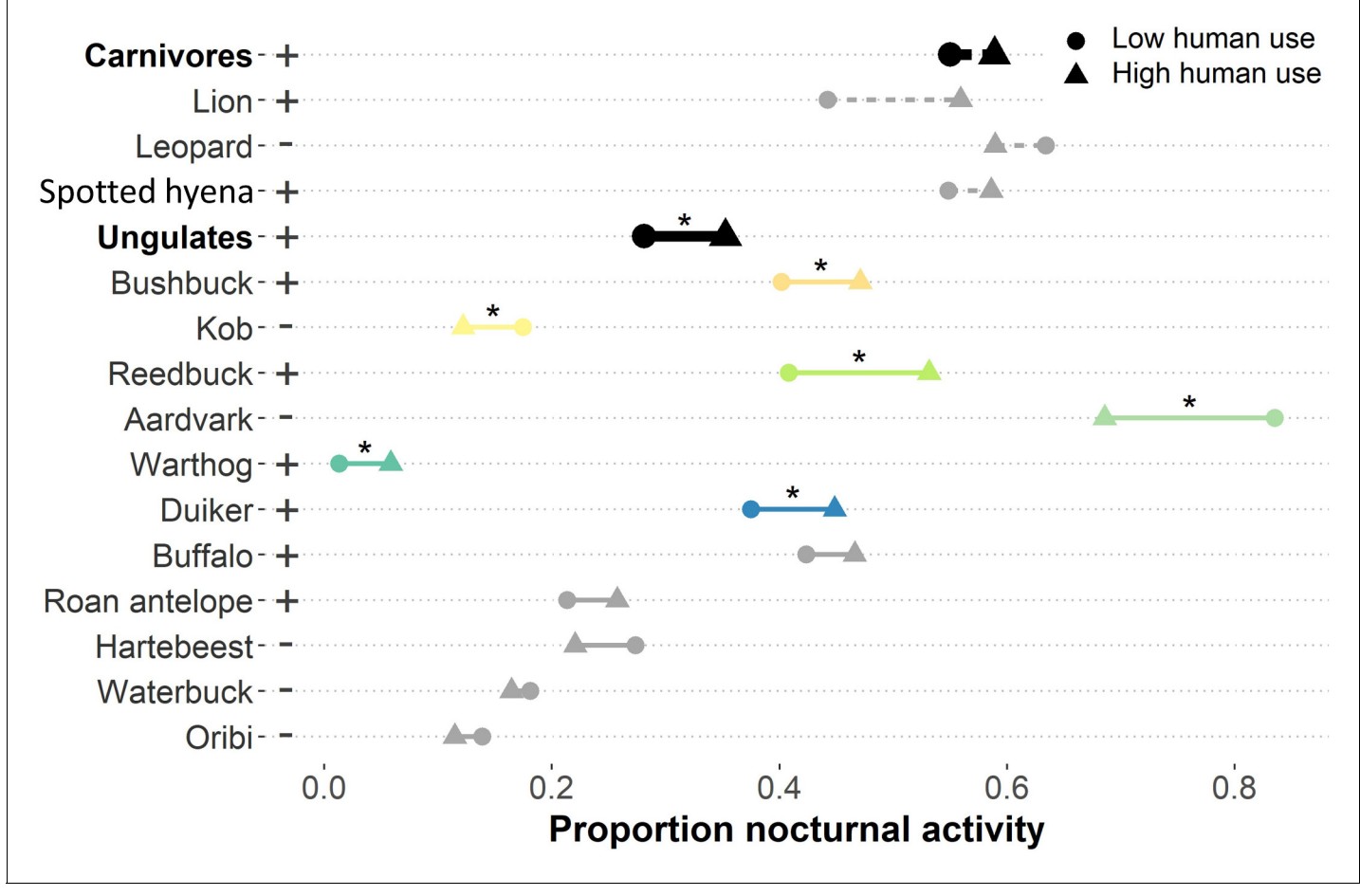

**Figure 4.** Proportion of activity during the nocturnal diel period (2 hr after sunset to 2 hr before sunrise) between low and high human zones for large carnivores (dashed lines) and ungulates (solid lines). Increases and decreases in nocturnality from low to high human use areas are indicated by plus (+) and minus (–) labels next to species' names, respectively. Stars (*) above colored lines indicate species that showed significant differences in nocturnal activity between human zones based on bootstrapped 95% CIs of nocturnality, and the colors of those species' lines correspond to species colors used in **Figure 5**.

(Fligner-Killeen test, p-value=0.03), indicating lower diversity of available prey and therefore more access to certain prey species compared to others where human presence was high (**Figure 5b**). African leopards and spotted hyenas showed no significant differences in access variability as a response to humans.

## Discussion

Wildlife responses to human activities have the potential to reshape natural ecological processes and trophic dynamics (**Hebblewhite et al., 2005**; **Dorresteijn et al., 2015**; **Suraci et al., 2019a**). When anthropogenic pressures are heterogeneous, the resultant dynamism promotes many adaptive strategies to manage and mitigate threats including behavioral shifts in diel activity that redefine species' temporal niches (**Muhly et al., 2011**; **Carter et al., 2012**; **Gaynor et al., 2018**; **Frey et al., 2020**). Such shifts in diel activity may lead to increased prey vulnerability to nocturnal predators, thus altering probabilities of encounter and diets in consumers (**Figure 1**). We found that over two-thirds of the assessed species exhibited different overall diel activity patterns as a response to humans in the study area. Most species showed more nocturnal activity, consistent with previous works and supporting our hypothesis of human avoidance (**Carter et al., 2012**; **Gaynor et al., 2018**). **Valeix et al., 2012** and **Suraci et al., 2019b** similarly found reduced diurnal activity near human settlements in African lions in Makgadikgadi Pans National Park, Botswana, and Laikipia,

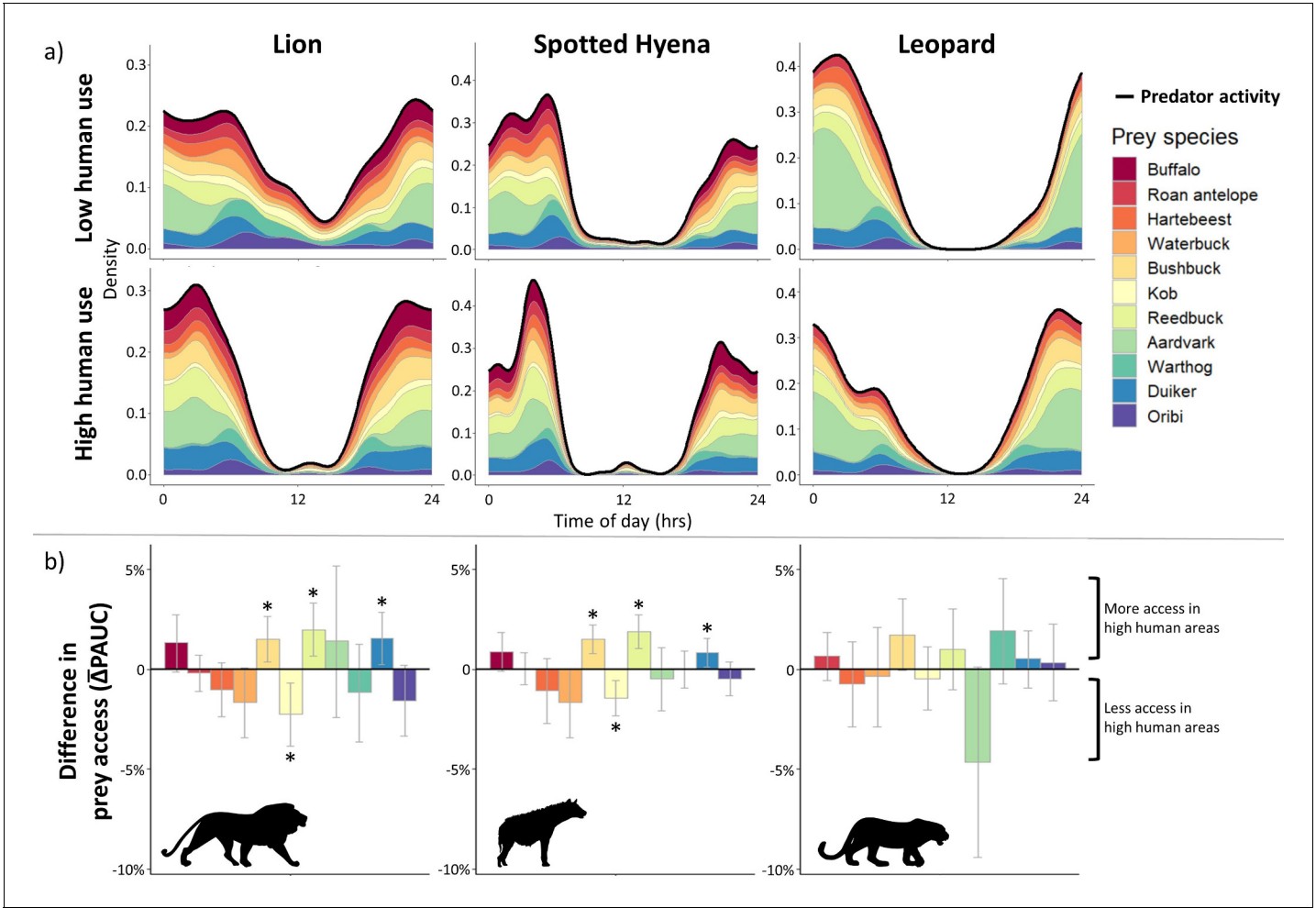

**Figure 5.** Differences in prey access between human activity zones for African lions, spotted hyenas, and African leopards from new temporally explicit community analysis. Buffalo was not included as prey for leopard. (**a**) Temporal overlap heatmaps representing the relative contributions of each prey species to the overall available prey base scaled to predator activity curve (solid black line) over the 24-hr cycle. Density values are calculated from kernel density temporal curves for predators and prey species. (**b**) Mean differences in prey access based on species-specific area under the predator activity curve (ΔPAUC) between areas of low and high human pressure, averaged among 10,000 parametric bootstrap replicates. Error bars represent bootstrapped 95% confidence intervals of $\overline{\Delta\text{PAUC}}$, with asterisks (*) designating significant differences in predator access to prey species.

Kenya, respectively, likely to reduce risks of human encounters. Human presence appears to be limiting temporal refugia from risks for many species and driving increases in ungulate activity when predators are also active, possibly decoupling anti-predator behaviors from predation risks (*Dröge et al., 2019*; *Smith et al., 2019*). *Patten et al., 2019* also presented evidence of human avoidance driving increased predation risks in North American white-tailed deer (*Odocoileus virginianus*).

Heterogeneity in species' responses to human presence, however, indicates different sensitivities to humans among the carnivores and ungulates in our study system. Some species did not exhibit differences in nocturnality as expected (e.g. kob and aardvark). These species may be benefitting from the observed human avoidance in many sympatric species that potentially reduces risks of predation and competition, commonly referred to as a human shield response (*Berger, 2007*; *Muhly et al., 2011*). For example, *Atickem et al., 2014* reported mountain nyala (*Tragelaphus buxtoni*) leveraging predator avoidance of humans during the day as a temporal refuge in Ethiopia. The ability to exploit human presence as a shield from predatory or competitive encounters may be due to the life history traits of a species that reduce sensitivity to humans, such as body size, energetic requirements, dispersal abilities, social structure, or foraging strategies (*Blumstein et al., 2005*;

Tablado and Jenni, 2017). Similarly, these species' temporal niches may be constrained by inherent characteristics that were evolved for diurnal activity, making night-time activity more costly despite refuge from human pressures and limiting their adaptive capacity to avoid humans (*Monterroso et al., 2013*). In contrast, the amount of wildlife persecution (i.e. trophy hunting and poaching) in the system may induce stronger human-avoidance behaviors in hunted species. For instance, *Vanthomme et al., 2013* attributed the negative associations of 10 mammals with human disturbances in Gabon to hunting avoidance behaviors, contrasted by six species in the study showing positive associations. Although the mechanisms driving differential responses to humans were not explicitly investigated here, our study demonstrates non-uniform responses of large mammals to human presence. As such, future work can assess the drivers of species-specific responses and sensitivities to humans.

We showed that human presence modified the availability of prey species relative to the overall pool of available prey, which is an important driver of prey selection in apex predators, and thus provide new insights into community-level repercussions of human sympatry with wildlife (*Sinclair et al., 2003*; *Owen-Smith and Mills, 2008*). While we expected overall predator-prey overlap and the diversity of available prey to be higher due to human avoidance, the combination of human avoidance and human shield strategies observed in our system resulted in little difference in overall overlap but substantial differences in apex predator access to individual prey species. Specifically, our new community-level approach to predator-prey temporal overlap revealed that prey species experienced intensified overlap with predators when they increased their nocturnal temporal niche (e.g. duiker, reedbuck, bushbuck) to avoid humans, while overlap was lessened for species that did not (e.g. kob). For African lions, this resulted in a lower diversity of available prey, likely intensifying predation pressures on a smaller subset of species which could contribute to destabilizing trophic dynamics (*Gross et al., 2009*). This highlights increasing concerns for the persistence of the now Critically Endangered West African lions that are suffering from prey depletion (*Henschel et al., 2014*). The predators in our study are largely opportunistic night-time hunters, and temporal overlap is often strongest between predators and their preferred prey species (*Hayward and Slotow, 2009*; *Linkie and Ridout, 2011*; *Ramesh et al., 2012*; *Dou et al., 2019*). Thus, we expect that species experiencing the highest overlap with apex predators relative to other prey to be integrated into the predators' diets in higher proportions, and consequently expect varied prey selection by predators between low and high human use areas. Buffalo are a common prey item of African lions in other systems, and our results suggest they may be vulnerable to intensified selection by lions due to human presence increasing access to buffalo in our study area (*Davidson et al., 2013*). Our approach implemented in this study may therefore be useful for anticipating herbivore population declines as a result of intensified predation pressures, as well as potential resulting feedbacks into predator population stability especially for endangered species such as the West African lion (*Owen-Smith et al., 2005*). Additionally, human disturbance can increase the predation rates and carcass abandonment by large carnivores as well as alter mesopredator foraging behaviors, potentially increasing mortality rates on preferred prey species and providing augmented carrion resources that may be detrimental to scavenger populations (*Smith et al., 2015*; *Prugh and Sivy, 2020*). As such, disturbances to predator-prey relationships potentially lead to alterations in predators' diets with consequences for ungulate and mesopredator community regulation and nutrient distribution (*Schmitz et al., 2010*; *Owen-Smith, 2019*).

Although protected areas are the primary strategy for biodiversity conservation worldwide, human exploitation of protected areas is pervasive and in many cases, necessary for the sustenance of human populations (*Jones et al., 2018*; *Geldmann et al., 2019*). By accounting for imperfect detection to understand human space use, we contribute to a more comprehensive understanding of human impacts within coupled human-natural ecosystems that is imperative to effectively manage for the conservation of ecological processes, biodiversity, and human needs. However, human activities observed in our study system may not impact species uniformly. Because we aggregated a variety of human activities to depict human use, there might be activity-specific responses by wildlife that were not captured. Humans exploit resources in national parks in many ways including livestock herding, resource gathering, subsistence poaching, hunting, and recreation, all of which impact the system and wildlife to varying degrees (*Everatt et al., 2019*; *Geldmann et al., 2019*; *Harris et al., 2019*). Indeed, *Harris et al., 2019* found differential impacts of human activities on wildlife behavior in WAP, suggesting species in this system do not respond to all humans uniformly. However, limited

sample sizes of many human activity categories currently preclude more detailed analyses using an occupancy framework. Overall, human impacts encompass a variety of disturbances that impact ecosystems, both in our study and more broadly, and thus disentangling the responses of wildlife to specific human pressures may facilitate designing more effective conservation interventions (*Jones et al., 2018*; *Nickel et al., 2020*). Our results are also suggestive of the potential ecological effects of changes to human activity in natural areas, which could result from fluctuations in tourism, infrastructure development, policy changes, and other local or global processes.

Our results demonstrate prevalent disruptions to wildlife temporal activity patterns from human presence, leading to overall reductions in diurnal activity and modified community dynamics. Because both carnivores and ungulates serve fundamental roles in regulating African ecosystems via predation and herbivory, respectively, the pervasiveness of their responses to human occurrence demonstrates the capacity for humans to disrupt essential ecological processes that facilitate coexistence among wildlife, in this case reshaping predator-prey interactions. As the human footprint continually expands, spatial refugia from anthropogenic disturbance become more limited, stimulating an increasing need to exploit temporal partitioning to avoid human pressures. We show that the community-level implications of these behavioral modifications must be considered in light of complex higher-order interactions that govern mechanisms of coexistence among predators and their prey.

## Materials and methods

### Study area

We conducted our study in the W-Arly-Pendjari (WAP) protected area complex that spans 26,515 km$^2$ in the transboundary region of Burkina Faso, Niger, and Benin (0°E-3° E, 10°N-13°N; *Figure 2a*). The complex contains 5 national parks (54% of total area), 14 hunting concessions (40%), and 1 faunal reserve (6%). Our study area within WAP comprised three national parks and 11 hunting concessions in Burkina Faso and Niger across ca. 13,100 km$^2$ (*Figure 2a*). Trophy hunting of many ungulate species and African lions (*Panthera leo*) is permitted in hunting concessions, while all hunting is illegal in the national parks and reserves in the complex. Other human activities in the park include livestock herding, resource extraction, recreation, and poaching (*Sogbohossou et al., 2011*; *Miller et al., 2015*; *Harris et al., 2019*). Recently, *Harris et al., 2019* reported 4 large carnivore species (African lion, African leopard *Panthera pardus*, spotted hyena *Crocuta crocuta*, and cheetah *Acinonyx jubatus*) and 17 ungulate species belonging to the superorder Ungulata in the three national parks included in our study area from an extensive camera trap survey. Cheetahs were detected only once, while wild dogs (*Lycaon pictus*) were not reported in the survey area. WAP has an arid climate and consists predominantly of Sudanian and Sahel savannas, with savanna accounting for ca. 90% of the habitat cover in the study area (*Lamarque, 2004*; *Mills et al., 2020*). We conducted our survey in the drier northern portion of WAP during the dry season with average monthly rainfall ranging from 0 to 1 mm in February to 42–91 mm in June (*Fick and Hijmans, 2017*). Although our study design may limit inferences to dry season conditions, evidence suggests that large African herbivores show similar overall temporal activity distributions as seasons change (*Owen-Smith et al., 2010*).

### Camera survey

We systematically deployed 238 white-flash and infrared motion-sensor cameras (Reconyx [Holmen, WI] PC800, PC850, PC900) within 10 × 10 km grid cells across our study area to assess effects of human presence on diel activity within the wildlife community. A single unbaited camera was placed within 2 km of the centroid in a total of 204 sampled grid cells over three survey seasons from January to June in 2016–2018 (*Figure 2—figure supplement 1*). Camera stations within cells that were surveyed in multiple years were not necessarily placed in the same location both years, but they were placed within the same 2 km buffer and are considered representative of the grid cell each year. Species identifications from camera images were validated by two members of the Applied Wildlife Ecology (AWE) Lab at the University of Michigan. We excluded false triggers, unidentifiable images, research team, and park staff from analyses. To ensure robustness in our analyses, we combined all remaining human images into a single 'Human' categorization representing a variety of human activities observed in WAP (e.g. livestock herding, resource gathering, recreation,

poaching, and hunting). Our work is not human subjects research requiring IRB review, although we remain grateful to authorities granting permission for our research and their efforts to manage coupled human-natural ecosystems. (see *Figure 2—figure supplement 1*, *Mills et al., 2020* and *Harris et al., 2019* for additional methods on camera deployment and image processing). Due to limited detections for some species, we aggregated survey data from all 3 years into a single dataset. We accounted for temporal variation in human space use during the subsequent modeling process, and previous work suggests little annual variation in wildlife activity (*Mills et al., 2020*). We created independence of species triggers using a 30-min quiet period between detection events using the 'camtrapR' package in R 3.5.1 (http://www.r-project.org) (*Niedballa et al., 2016*), and we assumed detections to be a random sample of each species' underlying activity distribution (*Linkie and Ridout, 2011*).

## Human occupancy models

We constructed single-season, single-species occupancy models to designate WAP into areas of low and high human use. We chose to use single-season models to assess the overall distribution of human space use across the study area, as opposed to multi-season occupancy models which also estimate extinction/colonization rates that is not necessary for our objectives. Instead, we included the year as covariate in single-species models to assess temporal variation in human occupancy patterns. Occupancy models account for spatial heterogeneity in human presence across the study area, facilitating investigation into the behavioral responses of sympatric wildlife. We separated detection/non-detection data for humans into 2-week observation periods, which were modeled as independent surveys to account for imperfect detection. Our occupancy models first modeled the detection process ($p$) using covariates expected to influence detection while holding occupancy ($\Psi$) constant, and then modeled human occupancy by incorporating grouping variables among which $\Psi$ may vary.

The global detection model included covariates related to survey design and the environment that we expected to influence the detection of humans: % savanna habitat (SAV), survey year (YR), trap-nights (TN), camera type (CAM), management type (MGMT), and site (i.e. one of 14 individual parks or concessions; SITE). MGMT was a binary variable that distinguished national parks from hunting concessions. Human occupancy was modeled with only grouping variables: MGMT, YR, and SITE. We included YR as a covariate to account for temporal variation in site use or detection, as cells surveyed in multiple years were considered separate sites for our single-season model. A grid cell surveyed in multiple years could, therefore, have different levels of occupancy between surveys. Variables included in the top performing occupancy and detection model(s) are considered those which best described the spatial variation in human detection and site use. We evaluated the support for all combinations of detection and occupancy covariates using the Akaike information criterion corrected for small sample sizes (AICc). We selected the top-performing detection and occupancy models as those with ΔAICc <2 compared to the lowest AICc model. We assessed goodness-of-fit of the top-performing models using 1000 parametric bootstraps of a $\chi^2$ test statistic appropriate for binary data and estimated the $\hat{c}$ statistic to ensure the data were not over-dispersed (*Fiske and Chandler, 2017*). We created all detection and occupancy models using the 'unmarked' package and conducted model selection using the 'MuMIn' package in R (*Fiske and Chandler, 2011*; *Bartoń, 2019*).

We extracted cell-specific latent occupancy probabilities, representing probabilities of site use by humans because the 10-km$^2$ grid cells do not meet the assumption of closure, from the top-performing (lowest AICc) occupancy model corrected for imperfect detection (*MacKenzie et al., 2016*). From those estimates, we categorized grid cells as either low or high human use. We delineated the threshold for human use using the mean value of human occupancy. We chose to use the mean occupancy as the threshold value because of the bimodal distribution of occupancy values and to facilitate comparisons between similar sample sizes of low and high human use grid cells (*Figure 2b*). We also conducted a sensitivity analysis to evaluate the selected threshold by repeating our analyses using thresholds ± 0.1, as described in the following section.

## Temporal analyses

Using detection timestamps from our camera survey, we compared the temporal activity patterns for apex predators (lions, leopards, and spotted hyenas) and sympatric ungulates between areas of low and high human use. We included 12 ungulate species: savanna buffalo (*Syncerus caffer brachyceros*), roan antelope (*Hippotragus equinus koba*), western hartebeest (*Alcelaphus buselaphus major*), waterbuck (*Kobus ellipsiprymnus defassa*), Buffon's kob (*Kobus kob kob*), Bohor reedbuck (*Redunca redunca*), bushbuck (*Tragelaphus sylvaticus*), aardvark (*Orycteropus afer*), warthog (*Phacochoerus africanus*), oribi (*Ourebia ourebi*), red-flanked duiker (*Cephalophus rufilatus),* and common duiker (*Sylvicapra grimmia*). We excluded four ungulate species from analysis in our study: topi (*Damaliscus korrigum jimela*) and red-fronted gazelle (*Eudorcas rufifrons*) due to few detections (<50), and elephant (*Loxodonta africana*) and hippopotamus (*Hippopotamus amphibius*) due to large body sizes that make them uncommon prey items for large carnivores. Duiker species were aggregated due to difficulty distinguishing the two in camera trap images, resulting in 11 total ungulate species in our analyses. Previous work in this system supports estimation of prey availability from camera trap data in that predator space use is heavily influenced by prey availability as estimated from camera trap detections (*Mills et al., 2020*). Further, temporal activity overlap between species directly influences the strength of interspecific interactions (*Kronfeld-Schor et al., 2017*).

We used kernel density estimation to produce diel activity curves representing a species' realized temporal niche in both human use zones for each of the 14 species. We first tested for differences in these activity distributions between low and high human use areas for all individual species and for each guild (representing the overall available prey base and predation pressures) by calculating the probability that two sets of circular observations come from the same distribution with a bootstrapped randomization test (*Ridout and Linkie, 2009*). Significant differences in temporal activities were evaluated as p-value<0.05. We conducted a sensitivity analysis by adjusting the human occupancy threshold ±0.1 and repeating this test for all species and both guilds to ensure robustness of our results (*Supplementary file 3*).

Using 10,000 parametric bootstraps of the temporal distribution models, we then calculated the area under the diel activity curves to determine the proportion of each species' activity that occurred during nocturnal hours (2 hr after sunset to 2 hr before sunrise). We used the sunrise (05:41) and sunset (18:06) times from the median date of our surveys (April 4, 2018) at the survey area centroid to define nocturnal hours. To test if wildlife nocturnality differed in response to human presence, we compared the bootstrapped 95% confidence intervals (CIs) of the difference in nocturnality for each species and overall guilds between low and high human areas where a significant difference was observed when the CI did not overlap 0.

We used the coefficient of overlap ($\Delta$) to quantify the total temporal overlap between each apex predator and their associated prey from circular activity distributions. Buffalo were excluded from the prey list of African leopards due to large body size. All other prey species were aggregated to produce a single diel activity curve of all prey for comparison to predator activity. We chose the specific estimator based on the minimum sample size of detections for both guilds to contrast human use levels ($\Delta_1$ if N < 75, $\Delta_4$ if N > 75). Values of $\Delta$ range from 0 to 1 where 0 represents no temporal overlap and 1 represents complete overlap or identical temporal niche between predators and their prey. We used 10,000 bootstrapped estimates to extract the bias-corrected 95% CIs of $\Delta$. We compared CIs of $\Delta$ between human use levels for each species to assess differences in predator-prey overlap in response to human occurrence. Non-overlapping CIs between human use levels indicated that the overall temporal overlap of predators with their prey was significantly altered by human presence. Temporal analyses were conducted using the 'activity' and 'overlap' packages in R (*Ridout and Linkie, 2009*; *Rowcliffe, 2019*).

## Predator access to prey

After determining overlap between predators and their prey as well as shifts induced by humans, we determined the implications for predator access to prey. To our knowledge, we developed a new method to assess species-specific prey access for predators that is temporally explicit over the diel period, enabling assessment of differences in the composition and diversity of accessible prey for predators resulting from responses to humans in both guilds. We first combined (i.e. stacked) the bootstrapped temporal kernel density curves for individual prey to produce a total diel activity curve

for prey, but this time maintaining each species' contributions to overall prey activity. We then multiplied each prey species' proportional contribution to prey activity at a given point in the diel cycle by the corresponding kernel density activity value of each respective apex predator. This method produced a discrete area under the predator temporal activity curve for each prey species of a given apex predator (percent area under curve, PAUC), where each prey species' value represents the relative temporal overlap between the apex predator and that prey species throughout the day. We used these PAUC values to assess whether predator access to individual prey species, relative to all available prey, were different between low and high human areas by calculating the difference in prey access (ΔPAUC) for each predator/prey combination between areas of low and high human use. To determine if prey access significantly differed between human presence levels, we compared bootstrapped estimates and 95% CIs of ΔPAUC. Finally, we used a Fligner-Killeen test for homogeneity of variance to determine how the diversity of each predator's accessible prey differed in association with human presence based on PAUC values. Lower variance in prey access represents more evenness (i.e. more diversity) in access across prey items, while higher variance indicates prey access is higher for a subset of species compared to others.

## Acknowledgements

We thank administrators in Ministries of Environment in Burkina Faso (OFINAP and DGEF) and Niger (DGE/EF) especially B Doamba and Y Harissou for logistical support as well as private concessionaires in Burkina Faso for wildlife management efforts and access to properties in WAP. We extend our sincerest appreciation to all park managers and the field team, especially I Gnoumou and YI Abdel-Nasser that assisted with data collection. We thank the AWE Lab for comments on manuscript drafts, assistance with image identification, and data management specifically S Gámez, M Lyons, J VanZoeren, and R Malhotra. We also thank B Dantzer and J Suraci for their valuable comments on the manuscript. We also acknowledge the University of Michigan (UM) African Studies Center - STEM initiative, UM Office of Research, the German Society of Mammalian Biology, and the Detroit Zoological Society for financial support.

## Additional information

### Funding

| Funder | Grant reference number | Author |
| --- | --- | --- |
| Detroit Zoological Society | | Nyeema C Harris |
| University of Michigan | Office of Research | Nyeema C Harris |
| Department of Afroamerican and African Studies, University of Michigan | STEM Initiative | Nyeema C Harris |
| German Society of Mammalian Biology | | Nyeema C Harris |

The funders had no role in study design, data collection and interpretation, or the decision to submit the work for publication.

### Author contributions

Kirby L Mills, Conceptualization, Data curation, Formal analysis, Investigation, Visualization, Methodology, Writing - original draft, Writing - review and editing, Processed and curated camera trap images; Nyeema C Harris, Conceptualization, Resources, Data curation, Supervision, Funding acquisition, Investigation, Visualization, Project administration, Writing - review and editing, Processed and curated camera trap images

### Author ORCIDs

Kirby L Mills  https://orcid.org/0000-0001-7693-9629
Nyeema C Harris  https://orcid.org/0000-0001-5174-2205

Decision letter and Author response
Decision letter https://doi.org/10.7554/eLife.60690.sa1
Author response https://doi.org/10.7554/eLife.60690.sa2

## Additional files

### Supplementary files

• Source data 1. Detection histories using 30-minute quiet period for 3 large carnivores and 11 ungulate prey species in the W-Arly-Pendjari complex obtained from a systematic camera survey over 21,430 trap-nights (January-June 2016-2018).

• Supplementary file 1. Species detections (using 30 min quiet periods) during the camera survey and common diel period. Asterisks (*) indicate significant shifts in diel activity distributions due to human presence. Changes in nocturnality are depicted for species with significant increases (+) and decreases (-) in response to humans. Empty cells represent no significant changes.

• Supplementary file 2. Human occupancy model selection table of top models with ΔAICc < 2 derived from camera data collected over three survey seasons in the W-Arly-Pendjari complex, West Africa. Detection ($p$) and occupancy ($\psi$) were modeled using the following covariates: CAM = camera type, SAV = percent savanna, YR = survey year, SITE = survey site, MGMT = management type (national park or hunting concession).

• Supplementary file 3. Sensitivity analysis of species shifts in circular activity distributions, by adjusting the threshold value of human occupancy ±0.1 from the mean. P-values are given for tests on species shifts using each threshold value. Sig. indicates the observed significance of shifts using the mean threshold value (0.54): + < 0.1, *<0.05, **<0.01, ***<0.001. The number of significant results (p-value<0.05) using different threshold values is given, for which three indicates significance using all thresholds and 0 indicates no significance for any threshold.

• Transparent reporting form

### Data availability

Occupancy model data (human detection histories and model covariates) are available on Dryad. Wildlife detection data have been provided as a supporting file (Source Data 1).

The following previously published dataset was used:

| Author(s) | Year | Dataset title | Dataset URL | Database and Identifier |
|---|---|---|---|---|
| Mills KL, Harissou Y, Gnoumou IT, Abdel-Nasser YI, Doamba B, Harris NC | 2020 | Data from: Comparable space use by lions between hunting concessions and national parks in West Africa | https://doi.org/10.5061/dryad.r4xgxd28g | Dryad Digital Repository, 10.5061/dryad.r4xgxd28g |

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
