## [Decision Letter]

**Acceptance summary:**

This study examines differences in predator and prey temporal activity patterns in areas of high or low human use by combining occupancy models, temporal activity kernel density estimates, and other statistical tests to examine potential differences. Overall, the study is relevant to conservation efforts and understanding human-wildlife interactions. The work provides a new perspective of predator-prey temporal overlap from a community perspective that will be of broad interest to a wide range of researchers.

**Decision letter after peer review:**

Thank you for submitting your article "Humans disrupt access to prey for large African carnivores" for consideration by *eLife*. Your article has been reviewed by Christian Rutz as the Senior Editor, a Reviewing Editor, and two reviewers. The following individual involved in the review of your submission has agreed to reveal their identity: John Heydinger (Reviewer #1).

The reviewers have discussed their reviews with one another, and the Reviewing Editor has drafted this decision letter to help you prepare a revised submission.

The editors have judged your manuscript to be of interest, but as described below, feel that a major re-write is required before it can be published – we would therefore like to draw your attention to changes in our revision policy that we have made in response to COVID-19 (https://elifesciences.org/articles/57162). First, because many researchers have temporarily lost access to the labs, we will give authors as much time as they need to submit revised manuscripts. We are also offering, if you choose, to post the manuscript to bioRxiv (if it is not already there) along with this decision letter and a formal designation that the manuscript is "in revision at *eLife*". Please let us know if you would like to pursue this option. (If your work is more suitable for medRxiv, you will need to post the preprint yourself, as the mechanisms for us to do so are still in development.)

Summary:

This paper examines differences in predator and prey temporal activity patterns in areas of high or low human use by combining occupancy models, temporal activity kernel density estimates, and other statistical tests to examine potential differences. Overall, the paper is well-written and informative and the questions being asked are relevant to conservation efforts and understanding human-wildlife interactions. The work provides a new perspective of predator-prey temporal overlap from a community perspective that will be of broad interest to a wide range of researchers.

Essential revisions:

There are a few critical underlying issues that need to be addressed in a major re-write. These include a lack of clarity concerning how temporal and spatial dynamics interact, the methods for assessing and assigning human occupancy, and the need for greater environmental information.

1) Language around temporal and spatial dynamics should be more precise.

The Abstract reads: "We assessed human-induced changes to wildlife diel activity" but the paper is not about changes, it is about different levels of diel activity in grid cells with different levels of human occupancy. This may seem pedantic, but it is central to the claims of the paper. Changes in wildlife diel activity would take a baseline of diel activity within a grid cell and then examine the temporal differences following changes in human occupancy within that same grid cell. For example, if (using a two week period) a grid cell had two weeks during which human occupancy was 0.6 and that was followed by two weeks in which human occupancy was 0.2, and one assessed diel activity of wildlife in relation to those different levels of human occupancy within the same space, then you would be assessing changes in animal diel activity. The approach compares differences in diel activity between different geographic areas but then makes claims about changes in diel activity. Our question is: what has changed? You say that "over two-thirds of species altered their diel activity in response to human presence" but it would be more accurate to say that ungulate diel activity differed in areas of low versus high human occupancy. Based on the approach to human occupancy (lumping together observations in two-week periods) one could look at changes in human occupancy and then look at changing diel activity within the same grid cell. This would be interesting because you could examine if there was a lag-time in diel activity following changes in human occupancy; if so, what was this lag? The fourth paragraph of the Results section reads, "we evaluated the effects of human presence on diel activity…and consequential alterations to predator-prey relationships", save the word "alterations" (which assumes a baseline) this is what the paper does, not assess changes. The paper does not "test for shifts" but for differences.

This confusion around changes versus differences gets to a central problem with the paper's language, which reads as if low human occupancy is the norm and high human occupancy is a departure. This may not be intentional, but it is pervasive. The reader is left with the understanding that a certain level of human occupancy is 'natural' or perhaps desirable. However, hominids and wildlife have been coexisting in this region for tens of thousands of years. Without an historical baseline for human occupancy one cannot really assess 'change' -- only that different human occupancy levels yield different diel activities. To assess change you need to have a baseline. If low human occupancy is considered a baseline, the paper needs to tell us why.

Human activity levels may have changed significantly in your study area during the recent COVID-19 anthropause. Perhaps you can add a sentence or two how this, based on your present findings, may have affected animals' diel activity patterns?

2) Assessment of human occupancy.

It remains unclear how human occupancy was arrived at and whether the method was appropriate. There are a couple issues. (i) The paper says that all human images were combined into a single 'Human' categorization (subsection “Camera survey”). Was this done regardless of the number of people in an image? Surely there is a substantive difference between the presence of 30 people and 2 people for the occupancy model. (ii) We are not sure if a relative occupancy model is appropriate; it may be, but is it really useful to compare the relative human occupancy of a grid cell in the far south-western portion of the study area with a grid cell in the far north-eastern portion? (iii) The paper mentions that different human activities could have different effects on wildlife diel activity, but this may be so central to the argument that it warrants more than a passing mention. Was there really no difference in diel activity in hunting versus non-hunting grid cells? Because only a certain time of year was examined, is it possible to compare between hunting and non-hunting seasons? We were unconvinced that all human activities could be treated similarly. If there is literature on this it warrants a mention.

Please provide further details on how the human use camera data are organized in the occupancy model. Why did you use a single-season versus multi-season model? It looks like there could be differences in site locations between years, and/or sites only surveyed in one, two, or all three years. How was this all accounted for? Isn't it possible that, by using multi-season models, a site could have change from low human use to high human use or vice versa over the three years? Please also justify using mean value of human occupancy as the pressure threshold. Did you test other occupancies? What happens if one defines the pressure threshold occupancy >0.27? or based on median occupancy? Or mode? You shifted the threshold from 0.54 to 0.44 at one point (subsection “Human avoidance responses”) – why?

For the human use data, are humans only using this area during the day (i.e., are all the photos from the day)? We think that some of the activity, especially illegal activity such as poaching, would be taking place at night. Please present the percentage of photos during the day/night. Similarly, do you think that the camera traps provide a good idea on how much humans are using an area? Do you think this differs between areas where human presence is legal versus illegal? For example, those conducting illegal activities would be more likely to watch for cameras and avoid walking in front of them versus those conducting legal activities.

For both the human and wildlife data, what is the justification for grouping data from three years over 200 sites? Would there not be differences in either, based on season, year, or other factors? Particularly for the wildlife temporal data, would you expect there to be site differences in temporal patterns due to environmental factors, or perhaps seasonal differences due to rainfall or other factors? Please include justification for grouping all of these data together.

3) Environmental information.

Data collection was limited to the dry season without discussion about differences between dry and wet season. This does not undermine the approach, but would seem to limit the claims to the dry season (particularly pertinent if one discusses downstream issues of ecosystem function).

Prey accessibility: the study area lies along a transition zone between savannah and forest. You account for differences in environment when analyzing camera images, however, in terms of prey availability, is prey movement (as recorded by cameras) analogous to prey availability across different ecosystems within the study area? Related is the unexamined assumption that prey movement (as captured by cameras) is a robust proxy of prey availability. This may be supported by the literature, but if so, please make it explicit.

A map providing the different types of ecosystems would be helpful. It seems that human occupancy and wildlife activity would be influenced by ecosystem type, but maybe we are wrong. If ecosystem type had little influence, please say so.

4) Overall conclusions for management.

Can you address the management implications of these findings? What can be done differently by management to protect these species given the research findings from this study?

5) Ethics approval.

We could not find any mention in the manuscript that this work received institutional ethical approval, and noticed that you indicated in the online form that no human and animal subjects were used. While we appreciate that the study was observational in nature, the placement and servicing of camera traps may have caused disturbance to wildlife, and images were collected and processed of humans without their consent. Please provide details of ethical review and approval, as well as of any other relevant permit paperwork.

---

## [Author Response]

Summary:This paper examines differences in predator and prey temporal activity patterns in areas of high or low human use by combining occupancy models, temporal activity kernel density estimates, and other statistical tests to examine potential differences. Overall, the paper is well-written and informative and the questions being asked are relevant to conservation efforts and understanding human-wildlife interactions. The work provides a new perspective of predator-prey temporal overlap from a community perspective that will be of broad interest to a wide range of researchers.

Thank you very much for your review and positive feedback. We are very happy to hear that our work will interest readers.

Essential revisions:There are a few critical underlying issues that need to be addressed in a major re-write. These include a lack of clarity concerning how temporal and spatial dynamics interact, the methods for assessing and assigning human occupancy, and the need for greater environmental information.1) Language around temporal and spatial dynamics should be more precise.The Abstract reads: "We assessed human-induced changes to wildlife diel activity" but the paper is not about changes, it is about different levels of diel activity in grid cells with different levels of human occupancy. This may seem pedantic, but it is central to the claims of the paper. Changes in wildlife diel activity would take a baseline of diel activity within a grid cell and then examine the temporal differences following changes in human occupancy within that same grid cell. For example, if (using a two week period) a grid cell had two weeks during which human occupancy was 0.6 and that was followed by two weeks in which human occupancy was 0.2, and one assessed diel activity of wildlife in relation to those different levels of human occupancy within the same space, then you would be assessing changes in animal diel activity. The approach compares differences in diel activity between different geographic areas but then makes claims about changes in diel activity. Our question is: what has changed? You say that "over two-thirds of species altered their diel activity in response to human presence" but it would be more accurate to say that ungulate diel activity differed in areas of low versus high human occupancy. Based on the approach to human occupancy (lumping together observations in two-week periods) one could look at changes in human occupancy and then look at changing diel activity within the same grid cell. This would be interesting because you could examine if there was a lag-time in diel activity following changes in human occupancy; if so, what was this lag? The fourth paragraph of the Results section reads, "we evaluated the effects of human presence on diel activity…and consequential alterations to predator-prey relationships", save the word "alterations" (which assumes a baseline) this is what the paper does, not assess changes. The paper does not "test for shifts" but for differences.This confusion around changes versus differences gets to a central problem with the paper's language, which reads as if low human occupancy is the norm and high human occupancy is a departure. This may not be intentional but it is pervasive. The reader is left with the understanding that a certain level of human occupancy is 'natural' or perhaps desirable. However, hominids and wildlife have been coexisting in this region for tens of thousands of years. Without an historical baseline for human occupancy one cannot really assess 'change' -- only that different human occupancy levels yield different diel activities. To assess change you need to have a baseline. If low human occupancy is considered a baseline, the paper needs to tell us why.

We have modified the language throughout the manuscript to reflect the nature of our findings more accurately, as requested. We have changed any instances where we previously implied humans caused changes in diel activity and replaced those with discussion of differences in diel activity between areas of high and low activity.

We agree that comparing changes in human occupancy and associated changes in wildlife diel activity within each grid cell would be a compelling analysis. However, we would face challenges with obtaining adequate sample sizes for each species per grid cell per year combination to assess those changes. For example, African lions were only observed in multiple observation periods in 10 grid cells, and we would only be able to assess changes if human occupancy also changed within those same 10 grid cells. To maintain the strength of our findings, we have decided not to alter our analytical approach and instead revise the language within the paper to address the reviewers’ concerns. We will be returning to the system for camera surveys during the 2021-2022 field season and look forward to incorporating these analyses after obtaining more data.

These changes can been seen in the Abstract, Introduction, Results section, Discussion section and Materials and methods section.

Human activity levels may have changed significantly in your study area during the recent COVID-19 anthropause. Perhaps you can add a sentence or two how this, based on your present findings, may have affected animals' diel activity patterns?

Unfortunately, we do not have the data available to analyze if any changes in human activity occurred due to the COVID pandemic, as our research group was unable to complete a camera survey in this system for the 2020 season. So, an analysis of the effects of the COVID pandemic on human presence in the system is not feasible at this time. Given the relevance of human activity changes in the system to this study, we have added the following considerations to the Discussion section: “Our results are also suggestive of the potential ecological effects of changes to human activity in natural areas, which could result from fluctuations in tourism, infrastructure development, policy changes, and other local or global processes.”

2) Assessment of human occupancy.It remains unclear how human occupancy was arrived at and whether the method was appropriate. There are a couple issues. (i) The paper says that all human images were combined into a single 'Human' categorization (subsection “Camera survey”). Was this done regardless of the number of people in an image? Surely there is a substantive difference between the presence of 30 people and 2 people for the occupancy model. (ii) We are not sure if a relative occupancy model is appropriate; it may be, but is it really useful to compare the relative human occupancy of a grid cell in the far south-western portion of the study area with a grid cell in the far north-eastern portion? (iii) The paper mentions that different human activities could have different effects on wildlife diel activity, but this may be so central to the argument that it warrants more than a passing mention. Was there really no difference in diel activity in hunting versus non-hunting grid cells? Because only a certain time of year was examined, is it possible to compare between hunting and non-hunting seasons? We were unconvinced that all human activities could be treated similarly. If there is literature on this it warrants a mention.

i) We did consider detection of humans regardless of the number of people appearing in the images because occupancy models consider only presence-absence data within an observation period, regardless of the number of individuals or detections that occur within that period. Also, the number of individuals observed in human detections did not exceed 8. Therefore, the occupancy estimates represent the probabilities that humans use a particular grid cell, as described in subsection “Human occupancy models”. The primary use of occupancy models here is to account for the imperfect detection of humans and gauge the probability that they used grid cells in which they were not directly observed. All grid cells where humans were observed on camera, therefore, had occupancies of 1 because humans did use that site. We understand that the term “intensity of use” may have led to confusion surrounding this interpretation and have removed that phrase from the manuscript in the 2 places it appeared (Discussion section and subsection “Human occupancy models”).

ii) Our study was designed to capture the heterogeneity in human presence across the study area in WAP in order to understand human impacts on sympatric wildlife behavior. Our occupancy models account for spatial variation in human presence, facilitating comparisons of human use among grid cells throughout the study area. We have added the following sentence to subsection “Human occupancy models” for clarification: “Occupancy models account for spatial heterogeneity in human presence across the study area, facilitating investigation into the behavioral responses of sympatric wildlife.”

iii) Regarding the responses of wildlife to different human activities, we have added the following to the Discussion section: “Indeed, Harris et al., (2019) found differential impacts of human activities on wildlife detections rates in WAP, suggesting species in this system do not respond to all humans uniformly. However, limited sample sizes of many human activity categories currently preclude more detailed analyses using an occupancy framework.” Regarding hunting pressures specifically, our research group does not currently have access to data surrounding hunting rates on species in the study area and so any questions surrounding sensitivities to hunting would be highly speculative. Also, the survey period took place during the legal hunting season, preventing any comparisons between hunting and non-hunting seasons. Recent work from these camera surveys in the NPs of WAP indicated that hunting was the least common direct human activity, so we do not have evidence of heavy poaching pressure in the study area (Harris et al., 2019). While we do expect that some human activities may be more impactful than others, we do not think that this detracts from the broader patterns of wildlife activity responses to overall human activity that are the result of this study. Similarly, it is common practice to include an aggregated metric of human pressures as evidence in many rigorous ecological studies (e.g., Rich et al., 2016; Suraci et al., 2019; Creel et al., 2019) because, if significant, that can lead to further scrutiny to discriminate relative impacts by type.

Please provide further details on how the human use camera data are organized in the occupancy model. Why did you use a single-season versus multi-season model? It looks like there could be differences in site locations between years, and/or sites only surveyed in one, two, or all three years. How was this all accounted for? Isn't it possible that, by using multi-season models, a site could have change from low human use to high human use or vice versa over the three years? Please also justify using mean value of human occupancy as the pressure threshold. Did you test other occupancies? What happens if one defines the pressure threshold occupancy >0.27? or based on median occupancy? Or mode? You shifted the threshold from 0.54 to 0.44 at one point (subsection “Human avoidance responses”) – why?

We chose to use a single-season occupancy model because we were primarily concerned with the level of utilization of human throughout the study area, but less interested in the changes human use from year to year. We added the following to the Results section: “We chose to use single-season models to assess the overall distribution of human space use across the study area, as opposed to multi-season occupancy models which also estimate extinction/colonisation rates that is not necessary for our objective. Instead we included year as covariate in single species models to assess temporal variation in human occupancy patterns.”

We captured the potential for yearly changes in human use by including year as a covariate in the single-season model. Thus, the single-season model does allow for a given grid cell to change in occupancy level from one year to the next if it was surveyed twice, and so it is possible that a single grid cell be classified as high human use one year and low human use the next year. We have added the following to subsection “Human occupancy models” to increase clarity: “A grid cell surveyed in multiple years could, therefore, have different levels of occupancy between surveys.”

We chose to use the mean human occupancy value (0.54) as a threshold because of the bimodal distribution of human occupancy values (Figure 2), and to facilitate comparison of wildlife diel activity patterns between low and high human areas by allocating roughly equal numbers of grid cells to both categories. We have added the following sentences to subsection “Human occupancy models”: “We chose to use the mean occupancy as the threshold value because of the bimodal distribution of occupancy values and to facilitate comparisons between similar sample sizes of low and high human use grid cells (Figure 2B). We also conducted a sensitivity analysis to evaluate the selected threshold by repeating our analyses using thresholds ± 0.1, as described in the following section.”

We tested for sensitivity of our results to this selected threshold to ensure it was not biasing out results, which is mentioned in the Results section. We had added to this line for clarity: “After testing the sensitivity of our results to the human occupancy threshold selected, we found that increasing or decreasing the low vs. high human occupancy threshold by ±0.1 did not alter our interpretation of species’ differences in diel activity (Supplementary file 3).” This sensitivity testing information is also included in the Materials and methods section. We tested sensitivity over a range of thresholds 0.54 ± 0.1 because this appeared to be a reasonable range of threshold values based on the bimodal distribution of human occupancy estimates, and the consideration of maintaining sample sizes of grid cells in each human designation to facilitate comparisons. Likewise, using the median (0.67) rather than the mean (0.54) did not change our results and only changed the human use designation of 9 grid cells (4% of sample size) from high to low, though that is not explicitly included in the manuscript. Because the majority of human occupancy estimates occurred near 0 or 1, our choice of threshold does not appear to substantially impact our results.

For the human use data, are humans only using this area during the day (i.e., are all the photos from the day)? We think that some of the activity, especially illegal activity such as poaching, would be taking place at night. Please present the percentage of photos during the day/night. Similarly, do you think that the camera traps provide a good idea on how much humans are using an area? Do you think this differs between areas where human presence is legal versus illegal? For example, those conducting illegal activities would be more likely to watch for cameras and avoid walking in front of them versus those conducting legal activities.

The majority of human detections occurred during the day. We have added information on human diel use in the Results section: “Humans exhibited mostly diurnal activity with 80.3% of detections occurring between sunrise and sunset.” We have also added the diel activity curve from human detections in Figure 2C.

While some amount of illegal activity may account for the night-time detections, we expect that the much higher intensity of activity during the day would be driving the overall effects on wildlife diel patterns. It is also possible that illegal activities might be less likely to be caught on camera, though this has not been tested in our system, however, activities that are illegal (e.g., poaching) are illegal throughout the study area. The illicit nature of these activities makes monitoring efforts difficult, and camera surveys provide a systematic monitoring effort and are conducive to occupancy analysis, as completed here, that accounts for the imperfect detection of humans in the system.

For both the human and wildlife data, what is the justification for grouping data from three years over 200 sites? Would there not be differences in either, based on season, year, or other factors? Particularly for the wildlife temporal data, would you expect there to be site differences in temporal patterns due to environmental factors, or perhaps seasonal differences due to rainfall or other factors? Please include justification for grouping all of these data together.

We do not expect that annual variation of the human and wildlife data to impact the validity or interpretation of our results, because we explicitly examined differences in wildlife temporal activity in relation to human space use across the complex. We do account for any differences in human occupancy and detections over different years by including year as a covariate, as stated in subsection “Human occupancy models”. If there are environmental variables that are also affecting human use between years, our modeling framework would capture that variation in estimating human occupancy. Because we associate wildlife diel activity to those human use levels, we are also incorporating any indirect effects of the environment’s temporal variation on wildlife diel patterns. Further, it is largely accepted that the primary factors driving wildlife diel patterns are predation risks, thermoregulation, and foraging needs (Cozzi et al., 2012; Kohl et al., 2019; Veldhuis et al., 2020), as mentioned in the Introduction. Though climatic variables may vary from year to year and influence thermoregulatory needs and foraging availability for herbivores, we expect that any changes to diel patterns due to these variables would impact the entire system (e.g., drought) and so any changes in wildlife diel patterns would be observed in both low and high human areas and thus not bias our results.

Because our occupancy model selection results do indicate that year is important in human use variation, our estimates of human occupancy and designations of low and high human areas account for temporal variation in human use. Further, limited sample sizes for multiple species (for example, lions, leopards, hartebeest; see Supplementary file 1) compelled us to group the data from all three years. Finally, previous work that used a single-season occupancy model to estimate large carnivore space use in the study area showed no indication that survey year was important in explaining carnivore site use. We have included the following in the Materials and methods section to justify this decision: “Due to limited detections for some species, we aggregated survey data from all 3 years into a single data. We accounted for temporal variation in human space use during the subsequent modeling process, and previous work suggests little annual variation in wildlife activity (Mills et al., 2020).”

3) Environmental information.Data collection was limited to the dry season without discussion about differences between dry and wet season. This does not undermine the approach, but would seem to limit the claims to the dry season (particularly pertinent if one discusses downstream issues of ecosystem function).

We have added the following statement to subsection “Study area”: “Though our study design may limit inferences to dry season conditions, evidence suggests that large African herbivores show similar overall temporal activity distributions as seasons change (Owen-Smith et al., 2010).”

Prey accessibility: the study area lies along a transition zone between savannah and forest. You account for differences in environment when analyzing camera images, however, in terms of prey availability, is prey movement (as recorded by cameras) analogous to prey availability across different ecosystems within the study area? Related is the unexamined assumption that prey movement (as captured by cameras) is a robust proxy of prey availability. This may be supported by the literature, but if so, please make it explicit.

We have shown in a previous study that predator space use in this system is strongly associated with prey availability as measured by prey capture success at camera traps (Mills et al., 2020), supporting the use of camera data as a proxy for prey availability. Further, higher movement of wildlife inherently increases the likelihood of encounters with sympatric species and temporal overlap between species increases the strength of interspecific interactions. Using camera data to estimate prey availability for inference into predator-prey interactions has been well-established in the literature (e.g., Murphy et al., 2018, Burton et al., 2012). We have added the following justification in subsection “Temporal analyses”: “Previous work in this system supports estimation of prey availability from camera trap data in that predator space use is heavily influenced by prey availability as estimated from camera trap detections(Mills et al., 2020). Further, temporal activity overlap between species are directly influences the strength of interspecific interactions (Kronfeld-Schor et al., 2017).”

A map providing the different types of ecosystems would be helpful. It seems that human occupancy and wildlife activity would be influenced by ecosystem type, but maybe we are wrong. If ecosystem type had little influence, please say so.

The habitat of the study area is about 90% savanna habitat, with the little remaining area represented by mostly riparian forest (~6%), wetland/floodplain (~3%). There are no grid cells with less than 63% savanna habitat. We have also searched the literature and were unable to find any substantial evidence for differences in large mammal diel activity patterns between habitat types (see Owen-Smith et al., 2020). Because of the majority savanna habitat at the scale of 10-km grid cells and a lack of literature supporting differences in wildlife diel patterns between habitat types, we do not expect that our results are substantially influenced by habitat types.

We have added the following information to to the Discussion section: “…, with savanna accounting for ca. 90% of the habitat cover in the study area (Mills et al., 2020).”

4) Overall conclusions for management.Can you address the management implications of these findings? What can be done differently by management to protect these species given the research findings from this study?

We have added the following text to the Discussion section: “This highlights increasing concerns for the persistence of the now Critically Endangered West African lions that are suffering from prey depletion,”; and “Our approach implemented in this study may therefore be useful for anticipating herbivore population declines as a result of intensified predation pressures, as well as potential resulting feedbacks into predator population stability especially for endangered species such as the West African lion.”; and consideration of the implications of the COVID pandemic, as mentioned earlier,: “Our results are also suggestive of the potential ecological ramifications of the COVID-19 pandemic that substantially limited human presence in WAP following this study. This global reduction in human use of natural areas may lead to changes in predator-prey interactions and other fundamental ecological processes.”

5) Ethics approval.We could not find any mention in the manuscript that this work received institutional ethical approval, and noticed that you indicated in the online form that no human and animal subjects were used. While we appreciate that the study was observational in nature, the placement and servicing of camera traps may have caused disturbance to wildlife, and images were collected and processed of humans without their consent. Please provide details of ethical review and approval, as well as of any other relevant permit paperwork.

We consulted with the University of Michigan Institutional Review Board, which has determined that a full ethical review was not necessary for using human images in our study in part from working in public spaces with no expectation of privacy and government permit approving camera research. We have included the following statement in our Materials and methods section: “Our work is not human subjects research requiring IRB review, though we remain grateful to authorities granting permission for our research and their efforts to manage coupled human-natural ecosystems.”